# Retinal oxygen saturation changes progressively over time in diabetic retinopathy

Sveinn Hakon Hardarson[1]*, Einar Stefánsson[1,2], Toke Bek[3]

1 Institute of Physiology, Department of Medicine, University of Iceland, Reykjavik, Iceland, 2 Department of Ophthalmology, Landspitali University Hospital, Reykjavik, Iceland, 3 Department of Ophthalmology, Aarhus University Hospital, Aarhus N, Denmark

* sveinnha@hi.is

## Abstract

### Purpose

According to cross-sectional studies, oxygen saturation is elevated in retinal vessels in diabetic patients. We evaluated how retinal oxygenation (metabolic marker), vessel diameters and retinopathy grade (structural markers) change over time in diabetic patients.

### Design

Prospective cohort study following screening in a hospital setting.

### Methods

Retinal oximetry images were acquired in 214 patients with the Oxymap T1 oximeter. Imaging was repeated after a median of 3.0 years (range 0.76–6.8 years). Oxygen saturation and vessel diameters were measured in the right eye. Semiquantitative grading of retinopathy according to international guidelines and red lesion count were performed on fundus photographs.

### Results

Retinopathy grade according to the international semiquantitative grading system was unchanged. Arteriolar saturation increased by $0.75\pm0.15$ percentage points per year of follow-up ($p<0.0001$). Venular saturation increased by $1.74\pm0.26$ percentage points per year ($p<0.0001$) and arteriovenous difference decreased by $0.99\pm0.20$ percentage points per year ($p<0.0001$). Arteriolar diameters decreased by $2.7\pm8.5\mu m$ ($p<0.0001$) between visits and venular diameters decreased by $2.4\pm9.1\mu m$ ($p = 0.0002$). Median increase in red lesion count between visits was 2 lesions (range -128 to 212 lesions, $p<0.0001$). The change in red lesion count and change in diameters did not correlate with the length of follow-up ($p>0.44$).

**Data Availability Statement:** All relevant data are within the paper.

**Funding:** Icelandic Center for Research (ES, SHH) The Toyota Foundation (TB) University of Iceland Research Fund (ES, SHH) Helga Jonsdottir and

Sigurlidi Kristjansson Memorial Fund (ES, SHH) The funders had no role in study design, data collection and analysis, decision to publish, or preparation of the manuscript.

**Competing interests:** Sveinn Hakon Hardarson and Einar Stefánsson have commercial interest in the company Oxymap ehf. They have stock in the company, are on its board and are listed on two patents related to retinal oximetry (Automatic registration of images US 7774036 B2, Temporal oximeter WO 2010143208 A3). This does not alter our adherence to PLOS ONE policies on sharing data and materials

## Conclusions

Oxygen saturation in larger retinal vessels can increase and arteriovenous difference can decrease over time in diabetic patients without any observable changes in retinopathy grade. The results suggest that changes in retinal oxygen saturation may precede progression of diabetic retinopathy or that oxygen saturation is more sensitive to disease progression than retinopathy grade.

## Introduction

Diabetic retinopathy is diagnosed on the basis of morphological lesions in the retina related to disturbances in retinal blood flow, such as haemorrhages, microaneurysms, exudates, intraretinal microvascular abnormalities and neovascularizations [1, 2]. The new vessels develop secondary to capillary occlusion in the retinal periphery [3]. Hypoxia, which results from capillary occlusions, is assumed to stimulate the release of growth factors that initate the formation of new vessels to result in proliferative diabetic retinopathy [4]. The occlusion of retinal capillaries also stimulates the formation of shunt vessels to bypass the occluded retinal areas [5, 6].

It would be desirable to monitor diabetic retinopathy with parameters that reflect the early metabolic or ischemic stages of the disease rather than the consequent morphological changes. Monitoring metabolic changes, which are likely to occur early, may provide an opportunity for more timely management of the disease. Such a parameter might be the oxygen saturation in the larger retinal vessels that can be measured by dual wavelength retinal oximetry [7, 8]. Using this technique, the bypassing of the retinal capillary bed in diabetic retinopathy has been confirmed by an increased oxygen saturation in intraretinal shunts [6] and neovascularizations [9]. The resulting increase in oxygen saturation in retinal venules has been documented in a number of studies [10–21] although one study found a trend towards a decrease [22].

The evidence that oxygen saturation in larger retinal vessels increases with increasing severity of retinopathy is based on cross-sectional studies. In order to elucidate the clinical potential of retinal oximetry there is a need for prospective studies to document whether changes in retinal oxygen saturation precede or follow the development of the morphological lesions in the disease.

Therefore, changes in oxygen saturation over time were observed in a cohort of diabetic patients from which baseline data have previously been reported [14]. The possible correlation between changes in oxygen saturation in larger retinal vessels and changes in vessel diameters or the severity of retinopathy was tested.

## Methods

### Patients

All patients had given their written informed consent to participate in the examination programme and the collection of data for the project was approved by the local medical ethics committee.

The baseline examination had been performed on 722 consecutive patients examined at the Department of Ophthalmology, Aarhus University Hospital. The patients had been referred to the department from private ophthalmologists or the regional screening clinic for diabetic retinopathy for specialist evaluation of diabetic retinopathy. The clinical examination included fundus photography and retinal oximetry and is described in detail in [14]. In 294 of these

patients, the examination resulted in treatment of diabetic retinopathy and these patients were excluded from the study. In the remaining 428 patients, repeated oximetry was planned for subsequent follow-up examinations. Among these, 216 patients had died or had been referred for follow-up with a private practitioning ophthalmologist. Therefore, follow-up examinations that included oximetry were performed in 214 patients after a median time of 3.0 years (range 0.76 to 6.8 years). The background data of these patients at baseline are shown in Table 1.

## Retinal oximetry

Retinal oximetry images were acquired with the Oxymap T1 retinal oximeter (Oxymap ehf., Reykjavik, Iceland). The right eye of each patient was measured. The oximeter has been described previously [23]. It is based on a Topcon TRC-50DX fundus camera (Topcon Corporation, Tokyo, Japan). Beam splitting optics, narrow band-pass light filters and two digital cameras (Insight IN1800, Diagnostic Instruments Inc., Sterling Heights MI) are added to the top port of the fundus camera. The oximeter captures two images of the retina simultaneously at different wavelengths. One image is captured at 570nm and is insensitive to oxygen saturation and another image is captured at 600nm and is sensitive to saturation. The Oxymap Analyzer software finds retinal vessels in the image and estimates their light absorbance at both wavelengths. The light absorbance values are then used to estimate oxygen saturation in the retinal vessels.

## Data analysis

**Retinopathy grade.** The severity of retinopathy was graded on a semiquantitative scale with five steps ranging from no retinopathy to treatment requiring retinopathy according to international guidelines [1, 24]. The distribution of patients with different retinopathy grades on this scale was unbalanced, with all patients belonging to category 0, 1 and 2 [1]. Therefore retinopathy was also graded in more detail by an experienced grader who, blinded to the result of the oximetry, counted the number of red lesions (haemorrhages and/or microaneurysms) within a fovea-centered 60 degrees fundus photograph. All images were graded by an experienced nurse grader to whom the purpose of the study was unknown. The images were re-evaluated by a retinal specialist (TB), who agreed on all the grading levels determined by the nurse grader. None of the patients had clinically significant macular edema at baseline or follow-up.

**Retinal oxygen saturation.** At the time of the study, oximetry images from all follow-up examinations were anonymized and transferred to the University of Iceland, where the analysis was performed masked to grading of patients and other clinical data and the identity of the patients, using the Oxymap Analyzer software version 2.5.1 (revision 10981).

Oxygen saturation was calculated according to the formula

$$SatO_2 = (a \; x \; ODR + b) + (c \; x \; w + d),$$

**Table 1. Group characteristics at baseline (mean±standard deviation).**

| | |
|---|---|
| Gender (male / female) | 140 / 74 |
| Age | 55.3±15.2 years (n = 213) |
| Type of diabetes (type 1/ type 2) | 71 / 143 |
| Duration of diabetes | 16.8±11.0 years (n = 213) |
| Mean arterial blood pressure | 103.4±12.6 mmHg (n = 214) |
| HbA1c | 61.4±15.5 mmol/mol (n = 175) |
| BMI | 29.4 ±6.3 (n = 209) |

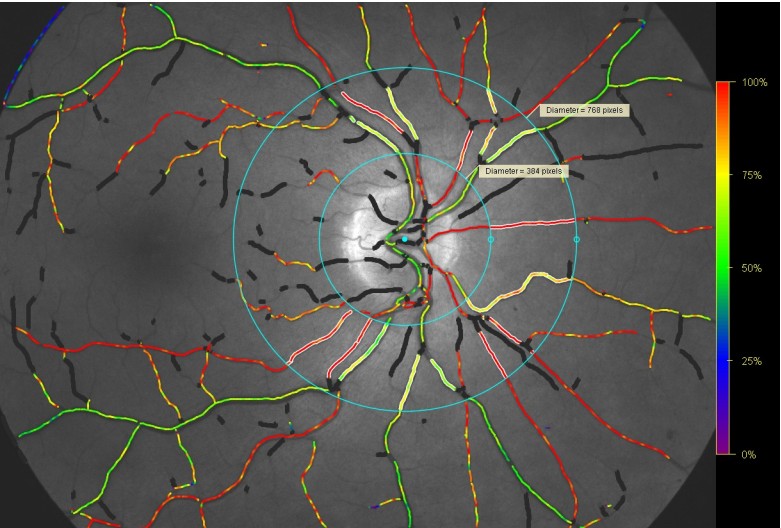

**Fig 1. An oximetry image.** The two blue circles demarcate the measurement area. The inner circle has a diameter of 1.5 disc diameter. The outer circle has a diameter of 3.0 disc diameters. The main retinal vessel segments in this area are chosen according to a set of detailed rules (Oxymap protocol from November 21st 2013, simple means used).

where a = -128, b = 124, c = 0.97 and d = -14. ODR is the optical density ratio, calculated from the images [23] and w is the width of the vessels in pixels.

A measurement area was demarcated with two concentric circles, with diameters of 1.5 and 3.0 disc diameters (see Fig 1). Segments of all retinal arterioles and venules, which had a diameter of 8 pixels (approx. 74 micrometres) or more [25], were measured between these circles. An average was calculated for all measured retinal arterioles and all measured retinal venules. The arteriovenous difference was calculated as the difference between those two numbers. This protocol for retinal oximetry is designed to measure (almost) all blood entering and leaving the retinal circulation. The oximetry results are therefore influenced by both the center and periphery of the retina.

Image quality score was automatically obtained for all images from the Oxymap Analyzer software and there was a significant (p = 0.002) reduction in the image quality from the first visit: mean = 7.7 (range 3.5 to 9.2) to the second visit: mean = 7.4 (range 4.8 to 9.1).

**Statistical analysis.** The oxygen saturation values were returned to Aarhus and were merged with the clinical data. Linear regression was used to test the following relationships: (1) The change in oxygen saturation with follow-up time. (2) The change in red lesion count with follow-up time. (3) The change in oxygen saturation with change in red lesion count. (4) The change in vessel diameter with follow-up time. (5) The change in oxygen saturation with change in vessel diameters. (6) The change in red lesion count with change in diameter. (7) The change in red lesion count with baseline HbA1c values or mean blood pressure.

Wilcoxon's signed rank test was used to test if the median change in red lesion count between visits was different from zero. A t-test was used to test if the change in vessel diameter between visits was different from zero.

Both mean and median values are given in the results when these differ considerably. For all statistical analyses, a parametric and non-parametric test were performed in parallel for completeness. Spearman's rank correlation and linear regression were performed on the same data. Wilcoxon's signed rank test or Mann-Whitney test and t-tests were performed on the

same data. In all but two analyses, the difference in results from a parametric and non-parametric test was negligible. In those two analyses, results from both tests are given.

Statistical analyses were performed with Graphpad Prism 5.01 (Graphpad Inc., San Diego CA).

## Results

Table 2 shows the retinopathy grades according to the International Classification [1] and the change in retinopathy grade during follow-up.

It can be inferred from Table 2 that retinopathy grade was unchanged in 83% of the patients, worsened by one grade in 10% and improved by one grade in 7% of the patients. The number of patients changing category was too low to allow conclusive testing against oximetry data. Therefore, red lesion count was used as a marker of retinopathy grade.

Fig 2 shows that A) arteriolar and B) venular saturation increased significantly with time of follow-up. The venular saturation increased more rapidly with time of follow-up, which implies that C) the arteriovenous difference decreased significantly with time of follow-up. Inner retinal oxygen extraction fraction, which is arteriovenous difference normalized by arteriolar saturation [26], also decreased significantly with time ($p < 0.0001$)

The overall red lesion count increased significantly by a median of 2 lesions, range -128 to 212 lesions ($p < 0.0001$, mean±SD: 15±47 lesions increase). However, Fig 3 shows no correlation between the change in red lesions and time of follow-up. Furthermore, the change in red lesion count was not correlated with baseline values of the known risk factors HbA1c ($p = 0.59$) or mean arterial blood pressure ($p = 0.55$).

The change in oxygen saturation over time was not correlated with the change in red lesion count (linear regression, $p = 0.83$ for arterioles, $p = 0.81$ for venules and $p = 0.63$ for arteriovenous difference).

During the follow-up period, the diameter of retinal arterioles decreased significantly by 0.29±0.91 pixels (approx. 2.7±8.5 micrometers, $p < 0.0001$) and the diameter of retinal venules decreased by 0.26±0.97 pixels (approx. 2.4±9.1 micrometers, $p = 0.0002$, median change = 1.30 micrometers). Neither the change in arteriolar nor venular diameter was correlated with the time of follow-up (linear regression, $p = 0.90$ and $p = 0.98$).

The change in diameter was in most cases correlated with change in saturation, see Table 3.

The change in diameter and the change in red lesion count were not correlated ($p = 0.29$ for arterioles change and $p = 0.50$ for venules).

## Discussion

Diabetic retinopathy is a slowly progressing disease, where it may require years or even decades to detect morphological progression [27]. Morphological changes are the end result of

**Table 2. Retinopathy grades at baseline and change in retinopathy grade during follow-up.**

| Baseline | Number of patients |
|---|---|
| Grade 0 (no retinopathy) | 24 |
| Grade 1 (mild non-proliferative retinopathy) | 54 |
| Grade 2 (moderate non-proliferative retinopathy) | 136 |
| **Change during follow-up** | |
| Grade 0→1 | 10 |
| Grade 1→2 | 12 |
| Grade 1→0 | 2 |
| Grade 2→1 | 13 |
| No change | 177 |

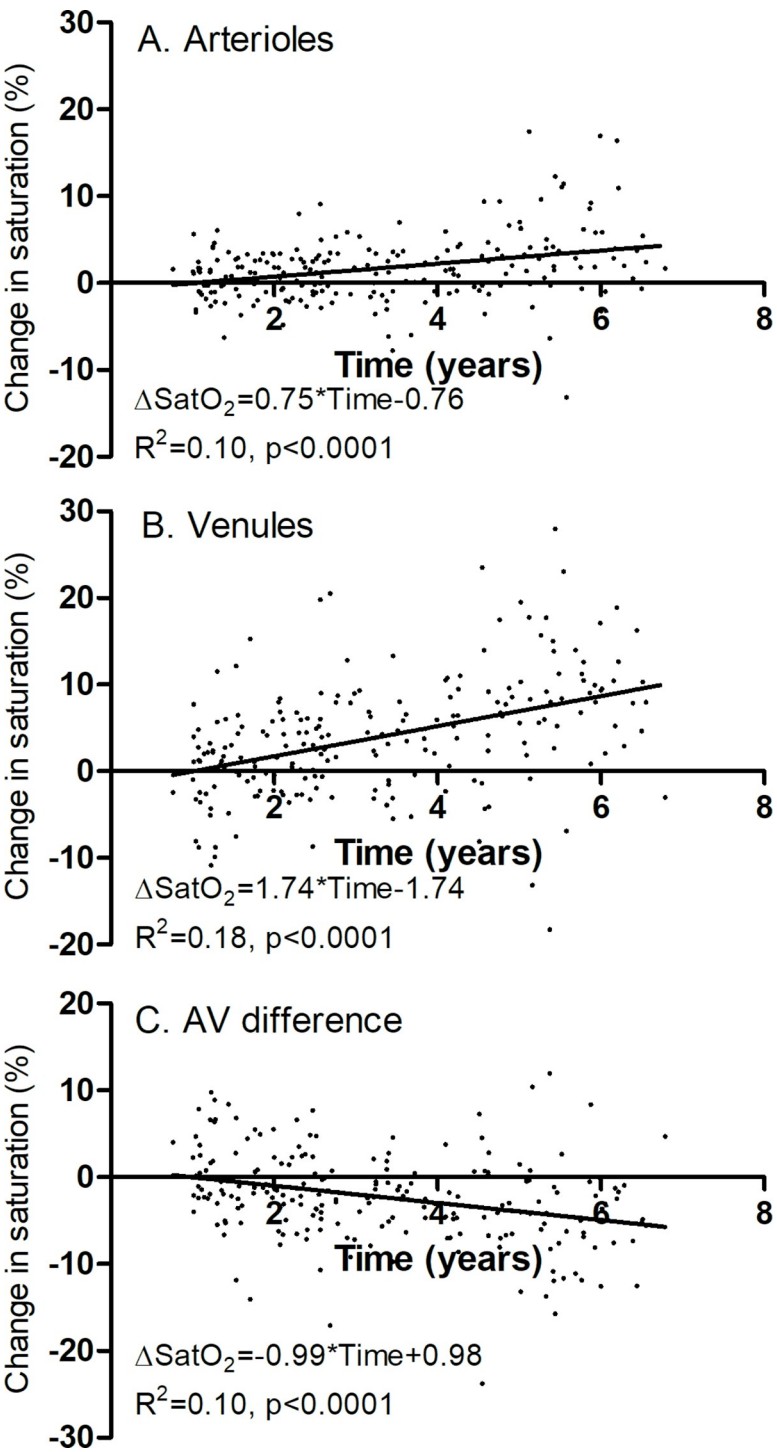

**Fig 2. The change in oxygen saturation.** Change from baseline to follow-up visit, plotted against the time of follow-up. Each point denotes one patient (n = 214). **A.** Saturation in retinal arterioles increased by 0.75 percent per year of follow-up (p<0.0001). **B.** Saturation in retinal venules increased by 1.74 percent per year of follow-up (p<0.0001). **C.** The arteriovenous difference in saturation decreased by 0.99 percent per year of follow-up (p<0.0001).

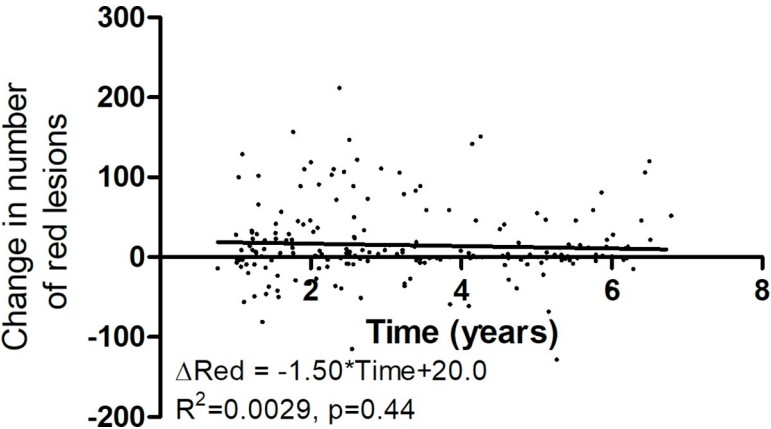

**Fig 3. The change in red lesion count.** Plotted as a function of follow-up time. There was no significant change in red lesions with time.

a chain of events that starts with metabolic dysregulation. The International classification of diabetic retinopathy is based on a semiquantitative scale that has few grades that precludes a detailed assessment of disease progression. Therefore, long follow-up periods are needed in order to study natural history or effects of interventions on the disease. The present study was conducted on patients referred to a secondary care center who had a higher risk for progression than the general population of diabetic patients. Therefore, it was considered relevant to study changes in oxygen saturation and retinopathy grade over a median follow-up time of 3 years [28, 29]. Nevertheless, there was little progression, which is likely due to the intensive control by endocrinologists for these patients and due to exclusion of patients who received ocular treatment between visits [30, 31]. Some patients had a very short follow-up time and others had longer. This variable follow-up time allowed us to test whether changes in oximetry values or structural parameters are correlated with length of follow-up.

Our results showed no clear progression of retinopathy, as defined by the International Classification [1]. Progression may have occurred within grades and could perhaps have been detected with more detailed measurements, such as OCTA. However, the standard grading is coarse and may, in many cases, have too little sensitivity to detect progression during this relatively short follow-up time. We therefore resorted to more detailed analysis of red lesion count for measuring morphological progression. We found a small overall increase in the red lesion count between visits but this was not related to the time of follow-up and there was a quite large variability in this parameter. This may be a consequence of fluctuations due to the fact

**Table 3. Linear regression between the change in saturation ($\Delta SatO_2$) between visits and change in diameter ($\Delta Diameter$).**

| | $R^2$ | P value for slope |
|---|---|---|
| $\Delta SatO_{2(arterioles)} = 0.83^*\Delta Diameter_{(arterioles)}+1.96$ | 0.037 | 0.0047 |
| $\Delta SatO_{2(arterioles)} = 0.99^*\Delta Diameter_{(venules)}+1.98$ | 0.061 | 0.0003 |
| $\Delta SatO_{2(venules)} = 1.69^*\Delta Diameter_{(arterioles)}+4.49$ | 0.050 | 0.0010 |
| $\Delta SatO_{2(venules)} = 1.05^*\Delta Diameter_{(venules)}+4.27$ | 0.022 | 0.028[a] |
| $\Delta SatO_{2(AV)} = -0.86^*\Delta Diameter_{(arterioles)}-2.53$ | 0.022 | 0.028[b] |
| $\Delta SatO_{2(AV)} = -0.06^*\Delta Diameter_{(venules)}-2.30$ | 0.00014 | 0.86 |

[a] p = 0.11 for non-parametric analysis.

[b] p = 0.13 for non-parametric analysis.

that a change in red lesion count reflects the balance between formation and resorption of lesions [32]. Altogether, the results indicate that with the present patient material a longer follow-up time is needed in order to identify changes in retinopathy grade.

Another structural parameter that we measured was the diameter of retinal vessels. There was a slight decrease in diameter of arterioles and venules between visits, which was significant for the group as a whole.

Oxygen saturation, however, clearly increased with length of follow-up in both arterioles and venules. The changes in saturation with time may reflect underlying metabolic progression of the disease that may later manifest more clearly in morphological progression of diabetic retinopathy. This will be tested with longer follow-up. It must be kept in mind, that most patients had retinopathy at baseline and it remains to be seen whether a change in oxygen saturation precedes the initiation of diabetic retinopathy.

Increased saturation in retinal arterioles has been observed in several cross-sectional studies [11, 12, 14–16, 18–21]. Various explanations have been proposed for this phenomenon, such as thickening of vessel walls and reduced counter-current exchange of oxygen between arterioles and nearby venules. Previous papers have also mentioned the possibility of faster retinal blood flow in diabetic patients, which would allow less oxygen to escape from per unit volume of blood through the walls of retinal arterioles [33]. The saturation increased more rapidly with follow-up in venules than in arterioles and the arteriovenous difference therefore decreased with follow-up. This result is in agreement with some of the cross-sectional oximetry studies [10, 12, 16, 17, 19], while other studies have not found such a decrease in the arteriovenous saturation difference [11, 18, 21]. A recent study showed that the linear velocity of the blood is inversely related to the measured oxygen saturation [34], and according to this, a decrease in this velocity might also contribute to the measurement of increased oxygen saturation in the larger retinal vessels. This would be in accordance with a gradual occlusion of the capillary bed in the periphery that is a prominent finding in the development of diabetic retinopathy [3, 35].

The inverse relation between linear velocity and measured oxygen saturation can be measured in both arterioles and venules and therefore the arteriovenous difference may be a more reliable parameter for oxygen saturations than arteriolar or venular saturation alone. A decreased arteriovenous difference in oxygen saturation may be a sign of poor delivery of oxygen by the retinal capillaries. This may be due to increased capillary occlusion and shunting to bypass the occluded vascular bed during the follow-up. A recent study has suggested that the arteriovenous difference may potentially be used as a measure of the extent of capillary occlusion [6]. This would imply that retinal oximetry could be used to quantify retinal changes that otherwise require invasive procedures such as fluorescein angiography or photographic techniques designed to capture images outside the central retina. This might point to a potential role of retinal oximetry for routine follow-up of diabetic patients.

The current study is the first longitudinal retinal oximetry study on diabetic retinopathy, apart from studies that included short-term follow-up to test the effect of treatment [13, 36–38]. Our previous cross-sectional study suggested that there is a relationship between oxygen saturation and severity of diabetic retinopathy [14]. However, oximetry and a combination of conventional risk factors were only able to explain 10 to 14% of the variability of the severity in that cross-sectional study. This may be due to the fact that diabetic retinopathy develops over a long time and measurements at a single point in time may not reflect the situation in the past years and decades, during which the disease has developed. In the current study, changes in red lesion count were quite variable between patients and not related to the time of follow-up. This may be due to fluctuations in risk factors, such as blood glucose and blood pressure or simply because clear structural changes take longer time to develop. Another factor that may

influence the results is the selection of patients. The selection process may mean that the studied sample is not entirely representative of progression of diabetic retinopathy in general. About two thirds of the patients were males and about two thirds of the group had type 2 diabetes. Most of the patients were of Caucasian origin. Importantly, we chose to exclude patients, who received ocular treatment between visits. This was done to isolate the effect of the disease progression, without confounding effects of different treatment. This probably also means that our selected patients had less progression than the study population as a whole.

The main strengths of retinal oximetry are its non-invasive nature and the ability to gauge metabolism. There are also limitations, which include possible effects of image quality on results and a relatively small size of change in individual patients. It is most likely that retinal oximetry can be used with other parameters, rather than as a standalone measurement. The same is true for almost every parameter in management of diabetes and diabetic retinopathy.

In conclusion, in a prospective study we have found progressive changes in retinal oxygen saturation in diabetic patients during a time interval of 3 years while no changes were detected on the conventional semiquantitative grading scale for diabetic retinopathy. This suggests that retinal oximetry is a sensitive measure of progression of the disease. While it may be assumed that diabetic retinopathy progresses over time, longer follow-up studies in cohorts where retinopathy grading actually increases are needed in order to confirm that the observed changes in oxygen saturation are followed by progression of retinopathy.

## Acknowledgments

The skillful assistance of nurse photographers Helle Hedegaard and Tina Bjerre is gratefully acknowledged.

## Author Contributions

**Conceptualization:** Sveinn Hakon Hardarson, Einar Stefánsson, Toke Bek.

**Data curation:** Sveinn Hakon Hardarson, Toke Bek.

**Formal analysis:** Sveinn Hakon Hardarson, Toke Bek.

**Funding acquisition:** Sveinn Hakon Hardarson, Einar Stefánsson, Toke Bek.

**Investigation:** Toke Bek.

**Methodology:** Sveinn Hakon Hardarson, Einar Stefánsson, Toke Bek.

**Project administration:** Einar Stefánsson, Toke Bek.

**Resources:** Toke Bek.

**Supervision:** Einar Stefánsson, Toke Bek.

**Validation:** Toke Bek.

**Visualization:** Sveinn Hakon Hardarson, Toke Bek.

**Writing – original draft:** Sveinn Hakon Hardarson.

**Writing – review & editing:** Sveinn Hakon Hardarson, Einar Stefánsson, Toke Bek.

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
