## [Decision Letter · Decision Letter 0]

18 Mar 2021

PONE-D-21-04724

Retinal oxygen saturation changes progressively over time in diabetic retinopathy

PLOS ONE

Dear Dr. Hardarson,

Thank you for submitting your manuscript to PLOS ONE. After careful consideration, we feel that it has merit but does not fully meet PLOS ONE’s publication criteria as it currently stands. Therefore, we invite you to submit a revised version of the manuscript that addresses the points raised during the review process.

The reviewers indicate that your study is worthwhile and presented well, but they list numerous minor shortcomings that should be addressable by modifying the text.  For example, if you analyzed only one eye per patient, how did you select that eye? What was the rate of progression in patients excluded because of treatment? Was the retinal grader aware of the grade assigned by the nurse grader?

We look forward to receiving your revised manuscript.

Kind regards,

Alfred S Lewin, Ph.D.

Academic Editor

PLOS ONE

Journal Requirements:

"Sveinn Hakon Hardarson and Einar Stefánsson have commercial interest in the company Oxymap ehf. They have stock in the company, are on its board and are listed on two patents related to retinal oximetry (Automatic registration of images US 7774036 B2, Temporal oximeter WO 2010143208 A3)."

4. Please ensure that you refer to Figures 1 and 2 in your text as, if accepted, production will need this reference to link the reader to the figure.

Reviewers' comments:

Reviewer's Responses to Questions

**Comments to the Author**

1. Is the manuscript technically sound, and do the data support the conclusions?

Reviewer #1: Yes

Reviewer #2: Yes

2. Has the statistical analysis been performed appropriately and rigorously? 

Reviewer #1: Yes

Reviewer #2: Yes

3. Have the authors made all data underlying the findings in their manuscript fully available?

Reviewer #1: Yes

Reviewer #2: Yes

4. Is the manuscript presented in an intelligible fashion and written in standard English?

Reviewer #1: Yes

Reviewer #2: Yes

5. Review Comments to the Author

Reviewer #1: Thank you for this interesting paper about the relationship between retinal oxygen saturation, DR level and red lesion count. It is a well conducted study with important results. However, I have some small corrections/suggestions to make the paper and especially the discussion section more in-depth and thorough. These are noted below.

Introduction:

Page 4, line 61-63: Can you please elaborate shortly why this would be desirable?

Methods:

Page 7-8, line 127-128: Was the retinal grader aware of the degree determined by the nurse grader when grading images? If yes, this could be subject to some bias. Please clarify this. Furthermore, add a section in the discussion section if this is the case.

Discussion:

Page 14, line 256-257: You may have a median follow up length of three years. However, there were a broad range in FU time. Please discuss what impact the difference in follow up length could have had on the results.

Page 15, line 278-280: How can you conclude this? Increase in oximetry is not a validated method for assessing progression of DR. Though it increased with time it is not completely clear that degree of DR does as well just because time has passed. It might be a stretch to conclude this. I suggest you rephrase, thank you.

Page 16, line 317: I suggest you also add a recently published paper in ACTA by Vergmann et al. (PMID: 33354935) that also involves this subject.

There was low quality in some images (down to 3.5) – is it possible to conduct proper analysis of images with that poor quality? Please comment on this, thank you.

What are your thoughts about the fact that you did not find any patients with level 3 or 4 DR? Please discuss this.

You mention a lot of strengths of oximetry. I suggest you also mention some of the limitations with this method.

Reviewer #2: 722 consecutive patients with diabetes underwent ocular examination including retinal examination, fundus photography and retinal oximetry with the Oxymap T1 retinal oximeter. The latter provides vascular diameter and oxygen saturation measurements. The examination resulted in treatment in 294 of these. Of the remaining 428 patients, 216 could not be followed-up. 214 patients were followed up for a median time of 3.0 years. The authors’ purpose was to relate the oximetry results to clinical diabetic retinopathy grades (as per Wilkinson, et al) and fundus red lesion counts. The authors rightly note the limitations of assessing diabetic retinopathy in terms of visible retinal lesions (Table 2 shows 83% patients had no change in clinical stage over the time studied). They also rightly show the need for physiologically based metrics, especially in when established longitudinally, as in the current manuscript.

The manuscript treats the results by patient, not eye. Apparently, only one eye was included in the study per patient. The manuscript should indicate how the eye included was chosen, eg all right eyes, the most severe eye, by randomization, etc. All patients had stage 0 to 2 (moderate nonproliferative retinopathy) and none progressed to stage 3 or higher. With a cohort this large over the time indicated, it would be surprising if none progressed to stages higher than stage 2. This lack of progression should be explained explicitly. Was it that as soon as that happened they were treated and excluded from the untreated cohort? Though not stated in the paper, it seems likely that any patient with CSME was treated and thus was excluded from the untreated cohort. It would be best to state whether any eyes developed CSME after the initial examination and, thus, were treated and then excluded.

It appears that gender and type of diabetes were not equally distributed, and it would be best to provide a statistical analysis to confirm this or disprove it. This would be mainly to be complete, since these factors are not likely to alter the results.

No statement on race is given. Elevation of retinal arterial SO2 in Hispanic patients was found in one paper (PMID: 29079858). It may well be that this cohort from Denmark had very few non-Caucasian patients, but that should be specified in the manuscript.

The key findings in the cohort were little change in clinical grade, no significant change in red lesions, increases in arterial and venous SO2, and decrease in arteriovenous SO2 over time. The latter indicates a greater increase in venous than arterial SO2. It would be of interest to report the arteriovenous SO2 difference normalized by the arterial SO2. This also equals the inner retinal oxygen extraction fraction. The largest oximetry change was in venous SO2 at 1.74 %/year. For oximetry to serve as a useful clinical tool to follow diabetic retinopathy, it would seem that measurements over a period of at least a year would be required.

A considerable limitation of the work is that a large number of patients could not be followed-up. Whether these progressed at the same rate as found in the cohort cannot be known. As someone who has participated in several longitudinal studies, this reviewer knows how difficult it is to minimize patient losses to follow-up. Still, this loss to follow-up should be cited as a limitation.

Of particular interest would be the rate of progression in the cases that were excluded because of treatment. As the paper reads, most of these were excluded at the first examination. However, there may have been some patients who progressed to treatable disease after having a few oximetry exams. This reviewer suspects that they may have been progressing faster than those in the cohort in the manuscript. If there were such patients, it would be of interest to include them. On the other hand, the authors may be collecting such cases for a future manuscript focusing on this group.

This reviewer notes that none of the papers from the Shahidi laboratory that have arterial and venous SO2 results in diabetic patients was cited in the Introduction. Inclusion of PMID: 29079858 and PMID: 27768785 is suggested.

6. PLOS authors have the option to publish the peer review history of their article (what does this mean?). If published, this will include your full peer review and any attached files.

Reviewer #1: No

Reviewer #2: No

---

## [Author Response · Author response to Decision Letter 0]

26 Apr 2021

Dear editors and reviewers

The response to the comments are included in an attached Word document.

---

## [Editor Report · Decision Letter 1]

29 Apr 2021

Retinal oxygen saturation changes progressively over time in diabetic retinopathy

PONE-D-21-04724R1

Dear Dr. Hardarson,

We’re pleased to inform you that your manuscript has been judged scientifically suitable for publication and will be formally accepted for publication once it meets all outstanding technical requirements.

Kind regards,

Alfred S Lewin, Ph.D.

Section Editor

PLOS ONE
---

## [Editor Report · Acceptance letter]

3 May 2021

PONE-D-21-04724R1 

Retinal oxygen saturation changes progressively over time in diabetic retinopathy 

Dear Dr. Hardarson:

I'm pleased to inform you that your manuscript has been deemed suitable for publication in PLOS ONE. Congratulations! Your manuscript is now with our production department. 

Kind regards, 

on behalf of

Dr. Alfred S Lewin 

Section Editor

PLOS ONE